# Predictive Power for Thrombus Detection after Atrial Appendage Closure: Machine Learning vs. Classical Methods

**DOI:** 10.3390/jpm12091413

**Published:** 2022-08-30

**Authors:** Pablo Antúnez-Muiños, Víctor Vicente-Palacios, Pablo Pérez-Sánchez, Jesús Sampedro-Gómez, Antonio Sánchez-Puente, Pedro Ignacio Dorado-Díaz, Luis Nombela-Franco, Pablo Salinas, Hipólito Gutiérrez-García, Ignacio Amat-Santos, Vicente Peral, Antonio Morcuende, Lluis Asmarats, Xavier Freixa, Ander Regueiro, Berenice Caneiro-Queija, Rodrigo Estevez-Loureiro, Josep Rodés-Cabau, Pedro Luis Sánchez, Ignacio Cruz-González

**Affiliations:** 1CIBERCV, University Hospital of Salamanca, 37007 Salamanca, Spain; 2Biomedical Research Institute of Salamanca (IBSAL), 37007 Salamanca, Spain; 3Philips Ibérica, 28050 Madrid, Spain; 4Instituto Cardiovascular, Hospital Clínico San Carlos, IdISSC, 28040 Madrid, Spain; 5CIBERCV, Instituto de Ciencias del Corazón (ICICOR), Hospital Clínico Universitario de Valladolid, 47003 Valladolid, Spain; 6Department of Cardiology, Health Research Institute of the Balearic Islands (IdISBa), Hospital Universitari Son Espases, 07120 Palma, Spain; 7Quebec Heart and Kung Institute, Laval University, Quebec City, QC G1V 0A6, Canada; 8Institut Clínic Cardiovascular, Hospital Clínic, Institut d’Investigacions Biomèdiques August Pi I Sunyer (IDIBAPS), 08036 Barcelona, Spain; 9University Hospital Alvaro Cunqueiro, 36312 Vigo, Spain

**Keywords:** left atrial appendage closure, device-related thrombosis, atrial fibrillation, machine learning, multivariable analysis, predictors

## Abstract

Device-related thrombus (DRT) after left atrial appendage (LAA) closure is infrequent but correlates with an increased risk of thromboembolism. Therefore, the search for DRT predictors is a topic of interest. In the literature, multivariable methods have been used achieving non-consistent results, and to the best of our knowledge, machine learning techniques have not been used yet for thrombus detection after LAA occlusion. Our aim is to compare both methodologies with respect to predictive power and the search for predictors of DRT. To this end, a multicenter study including 1150 patients who underwent LAA closure was analyzed. Two lines of experiments were performed: with and without resampling. Multivariate and machine learning methodologies were applied to both lines. Predictive power and the extracted predictors for all experiments were gathered. ROC curves of 0.5446 and 0.7974 were obtained for multivariate analysis and machine learning without resampling, respectively. However, the resampling experiment showed no significant difference between them (0.52 vs. 0.53 ROC AUC). A difference between the predictors selected was observed, with the multivariable methodology being more stable. These results question the validity of predictors reported in previous studies and demonstrate their disparity. Furthermore, none of the techniques analyzed is superior to the other for these data.

## 1. Introduction

Atrial fibrillation is the most prevalent arrhythmia worldwide, and more than one-third of the European population is expected to suffer from it in their lifetime [1]. This arrhythmia causes an inefficient auricular contraction, which increases the risk of thrombus formation, especially in the left atrium. It is worth noting that anticoagulation is indicated in those patients at high risk (CHA2DS2-VASC score ≥ 1 in men and ≥2 in women) to prevent thromboembolic events [2,3]. However, some patients are not suitable for this treatment, especially due to high risk of bleeding; therefore, percutaneous left atrial appendage occlusion has emerged as an alternative in the last decade [4,5]. The vast majority (>95%) of the thrombi secondary to atrial fibrillation originate in the left atrial appendage secondary to a reduce flow velocity of the blood inside it during atrial fibrillation [1]. In this procedure, the appendage closure is performed by deploying a device inside it from a transeptal approach from a femoral venous access [4]. Different devices have been created, and their design has improved throughout the last decade. Different studies have shown that excluding this appendage was non-inferior to oral anticoagulation to reduce the risk of thromboembolic events; thus, anticoagulation treatment can be withdrawn. Moreover, the bleeding risk can be reduced compared with oral anticoagulation [6,7].

The success rate of left atrial appendage occlusion is high with a low rate of complications, especially the vascular ones in the access site. This rate continues to increase due to new emerging materials and techniques. However, between 3–4% cases present device-related thrombosis during the follow-up. It consists in the formation of a thrombus in the auricular face of the device before correct epithelization; this is the reason why anticoagulation or dual antiplatelet therapy should not be withdrawn during the first months after the procedure [8]. This complication, although not frequent, confers a high risk of stroke and systemic embolism, so anticoagulation should be reintroduced [9,10]. Moreover, this is especially an issue in those patients at high risk of bleeding. On the other hand, DRT can occur not only during the first months after left atrial appendage closure but at any time during follow-up, so high-risk patients could benefit from a closer follow-up [11,12,13]. Therefore, it is crucial to predict those patients at high risk of device-related thrombosis.

Several authors have recently searched for predictors of thrombus detection after left atrial appendage closure with quite disparate results [9,10,11,13,14,15,16,17]. This divergence is accompanied by the difficulty of the task and the methodology used to find these predictors: multivariable regression [18].

Although this classical statistical methodology is widely used in the medical field, some misconceptions can lead to misinterpretation of results. One of them, and perhaps the most repeated, is not distinguishing multivariable from multivariate terminology [19]. Whereas multivariate analysis refers to statistical models with two or more dependent variables, multivariable analysis refers to statistical models with multiple independent variables.

Another major problem with these techniques is that they do not report on the goodness-of-fit of the model [20]. In a vast majority of cases, the models focus mainly on the significance of the variables and do not provide a value that assesses the goodness-of-fit of the regression. Nonetheless, the assumption of providing goodness-of-fit does not preclude the model from being generalizable.

One solution to provide a proof that our model generalizes correctly is through resampling techniques [21] either with an external sample or with bootstrapping techniques.

Combining multivariate regression with propensity score analysis is another widespread technique [22]. However, although these techniques attempt to alleviate possible causal problems, they do not solve the aforementioned problems and are not free of controversy [23].

An alternative to find predictors is to apply machine learning models. Recent interest exists in applying machine learning techniques to clinical research, and cardiology is no exception [24].

Most machine learning applications applied to medicine are focused on trying to enable the physician to spend more time with the patient by eliminating routine tasks such as semi-automated labeling clinical reports [25] or assisting in the segmentation of medical images [26]. It is worth noting that the most successful applications are those in which human–machine interaction is part of the system [27]. However, in the case of precision medicine, machine learning techniques may not be entirely clear and can be misleading [28]. An example is the multitude of publications on the prediction of COVID-19 in medical imaging that have appeared in recent years and do not yield clear results or predictors [29].

Another drawback of thrombus analysis after atrial appendage closure is its low prevalence. This factor greatly affects the results reported with both classical statistical and machine learning methods. As discussed above for classical statistical techniques, one of the approaches to this problem is through resampling techniques [30]. However, in the case of machine learning, boosting and bagging techniques can be added [31].

Finally, the clinical databases available for this type of analysis are usually relatively small (~10^3^ cases) compared to those where machine learning is most effective (~10^6^ cases). In this context, and by combining the treatment of non-large databases with resampling techniques, steps have been taken towards a consistent clinical machine learning methodology [32,33].

Our aim is to determine whether any differences exist with respect to the predictive power between the classical and the machine learning methodology to predict thrombus after atrial appendage closure. Secondarily, we want to extract the predictor variables of both methodologies and analyze their differences. To this end, we review in detail both methodologies, compare them with and without resampling techniques, and extract their predictors variables. Compared to previous studies, we report the goodness-of-fit of our models.

## 2. Materials and Methods

### 2.1. Data

A multicenter registry including 1150 consecutive patients who underwent left atrial appendage closure (LAAC) from 5 different international centers was utilized for this study [34,35,36]. Patient’s follow-up and antithrombotic treatment at discharge were conducted according to each center´s criteria. Both TEE and CT were performed at their respective centers to detect thrombus on the device during the first 3 months after left atrial appendage occlusion. The criteria used for the diagnosis were those proposed by Aminian et al. [13], defined as density on the left atrial aspect of the device not explained by imaging artifact or normal healing, visible in multiple planes, and in contact with the device. Patients without appropriately follow-up or with multiple missing data were excluded. Data were gathered retrospectively. Periprocedural variables of the registry were selected, including clinical and anatomic variables. However, the criteria for the diagnosis of device-related thrombosis were not universal for all the centers, and both transesophageal echocardiography and cardiac computerized tomography were used in the follow-up. Moreover, the diagnosis depended on each operator.

All data collection and the analysis fulfill the Declaration of Helsinki of 1975.

Other input variables such as clinical data have been already published in other studies [34,35,36,37].

Although both multivariable analysis and machine learning methodologies for predictor detection are widely known, it is important to highlight certain aspects to compare them. Both methodologies are described, and we detail the experiments performed.

### 2.2. Multivariable Analysis

The process for the selection of predictors is divided into two stages.

First, a univariate analysis of the analyzed variables is performed. In the case of continuous variables, one-way ANOVA or *t*-test is used. If the variables are categorical, the chi-square test is utilized. In the case of an ordinal variable, the Mann–Whitney U test is applied. Once each of the corresponding tests has been performed, the significant variables are selected (*p*-value < 0.05). This procedure represents a dimensional reduction process, especially in those cases where the number of variables is large.

Afterwards, these previously selected variables are incorporated into the multivariable regression model. Once the regression model is fitted, the significance of this set of variables is assessed. The final predictors are those that meet the chosen significance criterion (*p*-value < 0.05).

This process was performed on the entire database, and no resampling process was undertaken.

### 2.3. Machine Learning Feature Selection

Even though many methods for selecting predictors within machine learning exist, we chose one of the most widely used techniques: meta-transformer based on importance weights [38].

Meta-transformer based on importance weights works as follows: once the algorithm has been fitted, variables are selected based on their specific weight in the model. A threshold is set for this selection. Those variables that remain below this threshold are discarded from the model.

This process is always accompanied by either internal or external resampling depending on the characteristics of our data.

An independent external sample is ideal to demonstrate that our model is generalizable. However, in a clinical setting, it is not always possible either due to imbalanced data or size [39]. To alleviate this problem, we applied cross-validation. More specifically, we used k-folds with repetition [33] and shuffle split [40].

The k-fold procedure divides the main sample into *k* groups of samples or folds of equal size whenever possible. The algorithm is adjusted using the *k*-1 folds, and the fold that is left out is used for testing. This process is repeated *n* times with resampling.

The shuffle split method generates a defined number of independent splits of train and test data sets. The samples are first shuffled and then split into these subsets.

The suitability of the selected variables is made based on a chosen metric. Our chosen metric was the receiver operating characteristic (ROC AUC) curve.

### 2.4. Experiments Performed

To compare both methodologies, two lines of experiments were followed: multivariable and machine learning.

Logistic regression with regularization [41], random forest [42], and gradient boosting [43] were the machine learning algorithms used [44]. Logistic regression without regularization was utilized as the classic algorithm for the multivariable analysis.

Predictors were collected along with their associated values (*p*-values or weights). The area under the ROC (AUC ROC) curve, the specificity, and sensitivity of each experiment were also reported.

The cut-off point of the probability curve was determined by optimizing the ROC curve resulting from each experiment.

#### 2.4.1. No Resampling Experiments

Both predictor selection methodologies were applied to the entire dataset.

#### 2.4.2. Resampling Experiments

A 2-fold resampling technique with 5 repetitions was applied in addition to a shuffle split with a test size of 40% and 2 splits.

The selection of the resampling parameters is determined by the low prevalence of thrombus after atrial appendage closure. These parameters ensure a minimum number of cases of thrombus in each fold or split.

### 2.5. Experiments Design

Based on the two lines of research, 4 experiments were set up. Experiment I and II refer to research Section 2.4.1, while experiment III and IV refer to Section 2.4.2. Figure 1 shows the workflow of the 4 performed experiments.

### 2.6. Software

The programming language used in this article was Python, and the libraries used were scikit-learn [40], statsmodels [45], and Scipy [46].

The developed code is open source and can be consulted at https://github.com/IA-Cardiologia-husa/LAAC_Thrombus_detection_MLvsClassical (last accessed on 8 August 2022) [47].

## 3. Results

Out of the total of 1150 patients included in the registry, 813 were finally analyzed. Baseline characteristics of the patients are summarized in Table 1. Those patients with significant missing values or without adequate follow-up were excluded. Device-related thrombus was observed in 35 patients (4.31%).

Results are presented in two subsections: predictive power and predictor variables.

### 3.1. Predictive Power

Table 2 shows the results of the different experiments of ROC AUC, sensitivity, and specificity based on optimizing ROC AUC.

In experiment I, sensitivity and specificity are imbalanced. The lack of regularization in the fitted logistic regression explains this result. On the other hand, experiment II presents a perfect fit for the non-linear algorithms (gradient boosting and random forest). Nonetheless, experiment II model is clearly overfitted for all samples.

Experiments III and IV were performed using the kfold technique with K = 2 and 5 repetitions and shuffle split method with two splits and 40% of test size. In contrast to experiments I and II, experiments III and IV gathered the values of the metrics for each of their iterations. In experiment III, the results show a better performance for the k-fold resampling scenario. In experiment IV, the best result in terms of ROC AUC is obtained with k-fold resampling in combination with logistic regression. Furthermore, this combination presents the best sensitivity—specificity balance.

Subsequent analyses were performed considering only the k-fold resampling method and logistic regression algorithm. This combination obtained the most plausible characteristics for experiment II and the most suitable for experiments III and IV.

Figure 2 shows the analyzed metrics in each of the iterations. Even though no significant differences between the two experiments in terms of the mean values of the analyzed metrics were found, Figure 2 shows that experiment III is less variable than the experiment IV. Nevertheless, the low sensitivity obtained in experiment IV and its low AUC ROC proves a low predictive power.

### 3.2. Predictor Variables

Table 3 shows the significant variables after univariate and multivariable analysis (experiment I). Only 3 variables out of 76 were significant in the univariate test. Among these three variables, just one (HAS-BLED score) was found to be significant in the multivariate test.

Figure 3 shows the variables selected by the machine learning methodology on experiment II. This figure illustrates the positive or negative importance of each variable on the variable to be predicted.

Table 4 shows the number of occurrences the predictors were selected for experiments III and IV. In the case of experiment III, both the number of univariate and multivariable tests are shown.

Experiment III, similarly to experiment I, shows robustness in variable selection. In the case of experiment IV, only three variables were selected in each iteration (previous CAD, HAS-BLED, and pre-coumadin warfarin).

To analyze *p*-values (classical methodology) and weights (machine learning methodology) fluctuation due to resampling, a predictor variation rate for each variable was defined as described in Formula (1).
(1)$$predictor\;variation\;rate=\frac{V_{i}-V_{ED}}{\sigma }\;\left(i=iteration,\;ED=entire\;dataset\right)$$

*V* are *p*-values in the case of the classical methodology and weights for the machine learning methodology.

The numerator of Formula (1) is the difference between the value (*p*-value or weights) of each iteration of the resampling and the value obtained in the experiments performed on the entire database. The denominator is the standard deviation of the values collected for the predictor variable in the total number of iterations.

Figure 4 shows the predictor variational rate for the selected variables. To assess whether there were significant differences for the variance ratio between the two methodologies, a Mann–Whitney U test was performed, preceded by a Levene test. The Levene test was significant (*p*-value = 0.039), while the Mann–Whitney U test was not (*p*-value = 0.937).

## 4. Discussion

Device-related thrombus continues to be one of the main challenges in LAA closure. Device endothelization after the procedure is not well-known yet. Therefore, recent research has focused on DRT predictors identification and different treatment strategies after occlusion to prevent thrombosis. For instance, one prospective non-randomized study analyzed different antithrombotic treatments after LAAC and showed that half dose of direct oral anticoagulants (DOACs) can reduce DRT [48]. Furthermore, different prospective trials are currently ongoing trying to elucidate this question. On the other hand, new materials and better devices are being developed in order to reduce the thrombotic risk, specially the first months before the endothelization is completed.

Different studies in the literature have tried previously to find predictors of DRT. However, each registry found different predictors and always with a suboptimal power of prediction. Classical statistical analysis was used in all of them. Simard et al. found both major and minor risk factors [10]. The major ones were iatrogenic pericardial effusion and a hypercoagulable state. On the other hand, the minor ones were deep LAAC implant (over 10 mm from the pulmonary ridge), renal insufficiency, and AF other than paroxysmal. They found that the presence of one major factor or two minor factors would double the risk of DRT [10]. Dukkipati et al. found different predictors such as history of TIA or stroke, vascular disease, and the LAA diameter [11]. Even though permanent AF was the only one risk factor found in both registries, the predictive power was modest. Moreover, other believed predictors such as pericardial effusion CKD or vascular disease were included in our models. Conversely, CHA2DS2VASC score was defined as DRT in a Watchman 2.5 registry [16].

In our study, we used a multicenter registry of highly trained centers. The incidence was 4.31%, similar to that described in previous registries, which seems to be very representative of the current situation worldwide [10]. Although its frequency is low, the clinical repercussion of it remains a serious and relevant issue in LAA occlusion.

We found that there is no significant difference in the predictive power of thrombus after atrial appendage closure between the classical multivariable and machine learning methodology. In both cases, the predictive power was low. Therefore, device-related thrombus remains difficult to predict after LAAC.

On the other hand, differences between the predictor variables extracted by the two methodologies were found. Whereas in the case of the multivariable methodology, the predictors were mainly constant, the machine learning methodology utilized several variables to fit its models.

Differentiated results were obtained within the two lines of experiments (without and with resampling). On the one hand, the predictive power of the machine learning methodology (LR: 0.7974, RF: 1.0, GB: 1.0 ROC AUC) was clearly higher than that of the multivariable methodology (0.5446 ROC AUC). On the other hand, the machine learning methodology presented a more balanced trade-off between specificity and sensitivity than the multivariable methodology. Although the difference is remarkable, it may not be representative because both the fitting and the evaluation were performed on the entire database, which clearly demonstrates overfitting [49]. This overfitting is more obvious in the case of gradient boosting and random forest algorithms.

Experiments with resampling revealed that no predictive power differences between the techniques were found. This result evidences the overfitting of the machine learning model without resampling. Nonetheless, it is noteworthy that in both methods, the ROC AUC values are low, minimally exceeding those of a random model. We should also mention the low sensitivity value of both models, which corroborates that neither method is able to predict thrombus.

These unsatisfactory results do not allow the identification of patients at increased risk of thromboembolic events. This problem has a high relevance since most of these patients also have a high risk of hemorrhage. Furthermore, although most of these complications manifest themselves within the first few months after left atrial appendage occlusion, some individuals may develop thrombus beyond the first six months [11]. Better predictive power could allow physicians to perform longer and more intensive follow-up through TEE and CT evaluation in those patients at higher risk.

Albeit no differences were found in the predictive power, differences were observed in the predictive variables selected by the different methods. The multivariable methodology proved to be more robust when selecting predictor variables. In both the experiments with and without resampling, the prominent variable was the scoring system for assessing 1-year risk of major bleeding in patients with atrial fibrillation (HAS-BLED). In contrast, the machine learning methodology employed a broader set of variables among which it repeated only three in both experiments: previous coronary artery disease, previous anticoagulant intake, and HAS-BLED. Although this divergence of choice occurred, the prediction of thrombus was similar.

Our findings question first the advantages of machine learning techniques over classical ones [50] and, second, prove the divergence of predictors in previous studies [9,10,11,13,14,15,16]. Older age [9], history of stroke [9,11], permanent atrial fibrillation, vascular disease, left atrial appendage diameter, left atrial appendage diameter [11], smoking, sex [14], and LAA orifice width [13] were some of the predictors reported. Moreover, even a randomized trial has shown that antithrombotic treatment at discharge with half dose DOACs has shown to reduce significantly the incidence of device-relate thrombus, and none of them have reported it as a powerful predictor [48]. Among all the studies, only one performed bootstrapping [15], and in none of them was the predictive power reported.

As a main limitation, our study included only three machine learning algorithms and two resampling techniques. Additionally, hyperparameter tuning techniques were not employed. However, the main idea of the study was to compare methodologies (multivariable and machine learning). Therefore, the comparison of these with a common algorithm (logistic regression) is appropriate.

Anther limitation is the lack of a structured protocol including postprocedural antithrombotic treatment and scheduled follow-up. However, different protocols for postprocedural antithrombotic treatment have been reported in LAAC studies and without a formal consensus. Furthermore, patients undergoing LAAC are very heterogeneous, which means that each patient may benefit from a different discharge treatment. Some of them undergo LAAC because of new thromboembolic events despite correct oral anticoagulation. On the other hand, others are at high risk of bleeding or had had a relevant bleeding episode on oral anticoagulation, so continuing with anticoagulation is forbidden. Finally, despite the large number of patients, the retrospective nature of the study is also a limitation, and only new hypotheses can be made about the findings observed.

We have been thorough in performing different experiments to make them comparable. Furthermore, we have provided the rate of variation of predictors to contrast both variable selection techniques.

In this article, we used a structured multicenter hemodynamic database similar to those used in recent publications [9,10,11,13,14,15,16]. Our database presented a clear imbalance between cases with and without thrombus after appendage closure. Given these circumstances, future research would have to address the problem attempting to balance the cases and incorporate other sorts of variables into the studies (e.g., image or hematologic data). Additionally, if machine learning is not capable of predicting DRT, there is something we still do not know about DRT. That is why several studies are trying to predict them with heterogeneous results. However, using higher databases or other anatomic characteristics, previous device rejection such as stent restenosis, procedural issues, or hematologic coagulation disorders could offer more insights into both classical and machine learning models.

We firmly believe that in order to report a predictor, it is necessary first to perform a cross or external validation and second to report the predictive power. These two conditions are independent of the method used and are sufficient to ensure the validity of the predictors.

## 5. Conclusions

DRT after LAA closure is rare but correlates with an increased risk of thromboembolism. We analyzed a multicentric registry, and we found that there is no significant difference in the predictive power of thrombus after atrial appendage closure between the classical multivariable and machine learning methodology. In both cases, the predictive power was low. These results question the validity of predictors reported in previous studies and demonstrate their disparity. Furthermore, none of the techniques analyzed is superior to the other for these data. Nevertheless, machine learning models could achieve higher predicting power in larger registries.

## Figures and Tables

**Figure 1 jpm-12-01413-f001:**
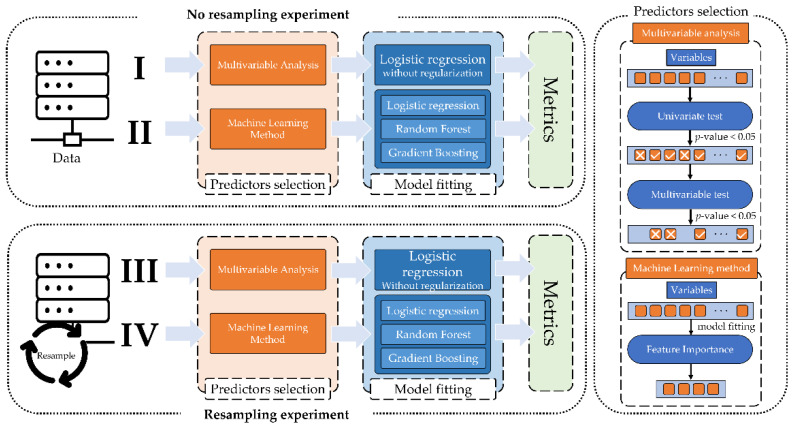
Experiment workflow.

**Figure 2 jpm-12-01413-f002:**
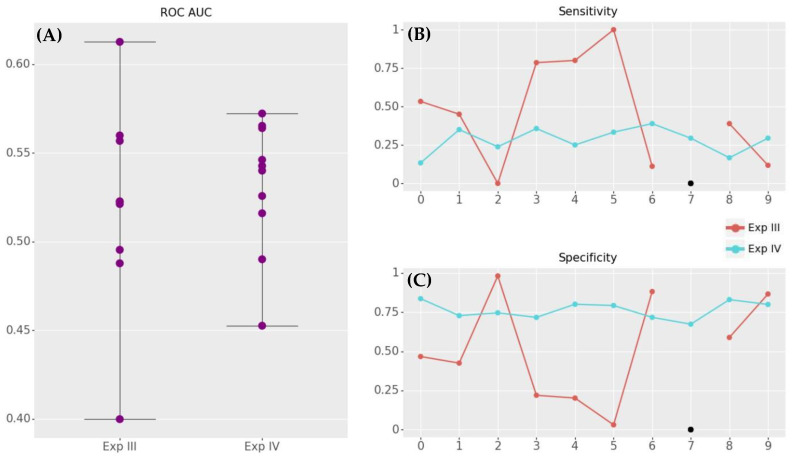
ROC AUC (**A**), sensitivity (**B**), and specificity (**C**) results for logistic regression in combination with k-fold resampling method (k = 2, 5 repetitions).

**Figure 3 jpm-12-01413-f003:**
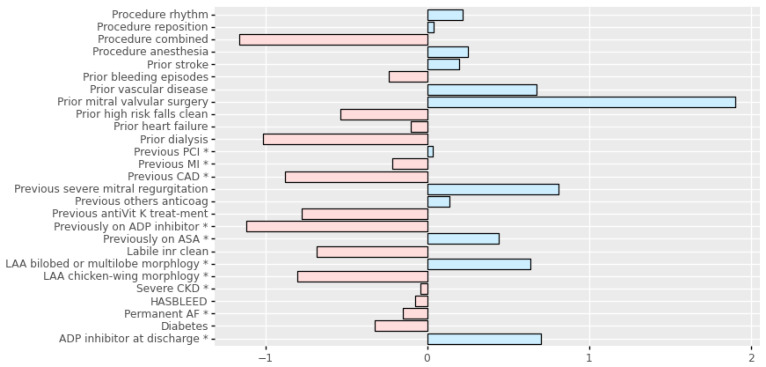
Variable weights results on experiment II using logistic regression. * AF, atrial fibrillation; ASA, acetylsalicylic acid; ADP, adenosine diphosphate receptor; CAD, coronary artery disease; CKD, chronic kidney disease; INR, international normalized ratio; LAA, left atrial appendage; MI, myocardial infarction; PCI, percutaneous coronary intervention.

**Figure 4 jpm-12-01413-f004:**
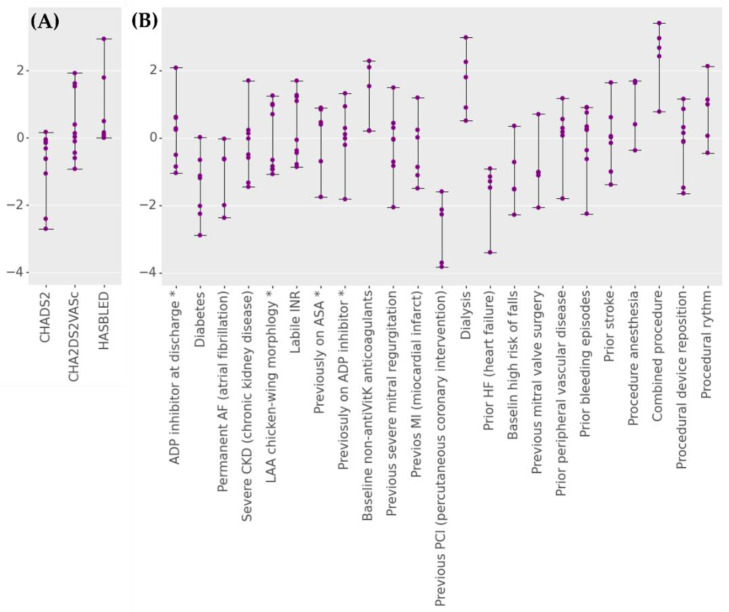
Predictor variation rate for each fold on the (**A**) experiment III and (**B**) experiment IV using logistic regression. Both experiments results used k-fold resampling method (k = 2, 5 repetitions). * ASA, acetylsalicylic acid; ADP, adenosine diphosphate receptor; LAA, left atrial appendage.

**Table 1 jpm-12-01413-t001:** Baseline Characteristics. Values are mean ± SD or % (*n*/*N*).

Variables	All Patients (*N* = 813)
Age (Years)	75.62 ± 8.43 (*813*)
Gender (Male)	61.01 (496/813)
Weight (Kg)	74.96 ± 15.55 (*806*)
Height (cm)	164.73 ± 9.41 (*804*)
BMI *	27.55 ± 4.94 (*804*)
BSA *	1.81 ± 0.21 (*804*)
HTA * (Yes)	87.33 (*710*/*813*)
Diabetes (Yes)	33.7 (*274*/*813*)
Smoking (Yes)	14.37 (*127*/*696*)
Prior stroke (Yes)	35.36 (*279*/*789*)
Prior hemorrhagic stroke (Yes)	24.8 (*185*/*746*)
Prior systemic embolization (Yes)	9.85 (*78*/*792*)
Prior vascular disease (Yes)	20.54 (*167*/*813*)
Previous CAD * (Yes)	31.89 (*258*/*809*)
Previous MI * (Yes)	20.08 (*154*/*767*)
Prior mitral valvular surgery	2.70 (*22*/*813*)
CHADS2	2.96 ± 1.24 (*768*)
CHA2DS2VASc	4.44 ± 1.54 (*813*)
HAS-BLED	3.71 ± 1.05 (*811*)
Prior bleeding episodes	1.57 ± 1.15 (*768*)
Labile INR * (Yes)	11.07 (*90*/*813*)
Previously on ASA * (Yes)	30.26 (*246*/*813*)
Previously on ADP inhibitor * (Yes)	11.07 (*90*/*813*)
Previous antiVitK treatment (Yes)	23.37 (*190*/*813*)
Previous severe mitral regurgitation (Yes)	15.23 (*122*/*801*)
Previous severe mitral stenosis (Yes)	0.26 (*2*/*765*)
Procedural device reposition (If available)	0.59 ± 1 (*570*)
Procedure contrast volume	134.03 ± 88.09 (*695*)
periprocedural pericardial effusion (Yes)	5.9 (*48*/*813*)
ADP inhibitor at discharge * (Yes)	62.64 (*508*/*811*)

* BMI, body mass index; BSA, body surface area; HTA, arterial hypertension; CAD, coronary artery disease; MI, myocardial infarction; INR, international normalized ratio; ASA, acetylsalicylic acid; ADP, adenosine diphosphate receptor.

**Table 2 jpm-12-01413-t002:** Results of the four experiments. The adjustment of the metrics was performed by optimizing the ROC AUC value.

Experiment	Resample	Feature Selection	Model	ROC AUC	Sensitivity	Specificity
I	-	Multivariable analysis	LR *	**0.5456**	**0.8857**	**0.2198**
II	-	Select from model	LR	0.7974	0.6857	0.7776
			RF	**1.0**	**1.0**	**1.0**
			GB	**1.0**	**1.0**	**1.0**
III	Shuffle split	Multivariable analysis	LR *	0.4387 ± 0.0904	0.1811 ± 0.1143	**0.6544 ± 0.2263**
	k-fold	Multivariable analysis	LR *	**0.5174 ± 0.0531**	**0.4318 ± 0.3255**	0.5504 ± 0.3173
IV	Shuffle split	Select from model	LR	0.4838 ± 0.0118	0.1697 ± 0.0724	0.6960 ± 0.0109
			RF	0.3989 ± 0.0240	0.0 ± 0.0	0.9014 ± 0.0639
			GB	0.4614 ± 0.0437	0.0444 ± 0.0459	**0.9217 ± 0.0073**
	k-fold	Select from model	LR	**0.5325 ± 0.0349**	**0.2804 ± 0.0771**	0.7634 ± 0.0518
			RF	0.4250 ± 0.354	0.0418 ± 0.0462	0.7049 ± 0.0193
			GB	0.4893 ± 0.0668	0.0754 ± 0.0501	0.9005 ± 0.0176

LR, logistic regression; RF, random forest; GB, gradient boosting. (*) LR without regularization.

**Table 3 jpm-12-01413-t003:** Experiment I predictor variables selection. Results from univariate and multivariable test on classic method.

Var	Univariate *p*-Value	Multivariable *p*-Value
CHADS2	<0.01	0.852
CHA2DS2-VASC	<0.01	0.414
HAS-BLED	<0.01	<0.01

**Table 4 jpm-12-01413-t004:** Occurrences of predictors for each feature selection method. Results for k-fold resampling method (k = 2, 5 repetitions).

Variable	Experiment III		Experiment IV
	Univariate	Multivariable	
Previous CAD *	-	-	10
HAS-BLED	10	8	10
Previous antiVitK treatment	-	-	10
CHADS2	10	-	6
CHA2DS2VASc	10	-	6
Chicken-wing LAA morphology	-	-	9
Previous severe mitral regurgitation	1	1	9
Prior bleeding episodes	-	-	9
Labile INR *	-	-	9
ADP inhibitor at discharge	-	-	8
Procedural device reposition	-	-	8
Severe CKD *	-	-	8
Periprocedural pericardial effusion	1	-	8
Diabetes	-	-	7
Prior stroke	-	-	7
Previously on ADP inhibitor *	-	-	7
Previously on ASA *	-	-	7
Prior systemic embolization	-	-	7
Prior vascular disease	-	-	6
Previous MI *	-	-	6
Prior hemorrhagic stroke	-	-	6
Procedure contrast volume	1	1	-
Prior valve surgery (mitral)	1	1	5

* CAD, coronary artery disease; INR, international normalized ratio; CKD, chronic kidney disease; ADP, adenosine diphosphate receptor; ASA, acetylsalicylic acid; MI, myocardial infarction.

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
