# Peer review of "Predictive Power for Thrombus Detection after Atrial Appendage Closure: Machine Learning vs. Classical Methods"

_jpm, 2022, doi:10.3390/jpm12091413_

Round 1

Reviewer 1 Report

The study is very interesting. However it should be expanded.

The authors should include several ML methods in the study before stating that: "Our findings question first, the advantages of machine learning techniques over classical ones". This can not be stated based on the results of only one, or two, ML methods.

The main limitation of the study should be resolved before publication.

Besides, if possible, authors could add some other input variables, such as: clinical data.

Author Response

  • English language and style are fine/minor spell check required
  • Is the research design appropriate? Must be improved
  • Are the methods adequately described? Can be improved

English language has been checked. Further explanation and changes about the research design and methods have been included in the new version

Comments and Suggestions for Authors

The study is very interesting. However, it should be expanded.

The authors should include several ML methods in the study before stating that: "Our findings question first, the advantages of machine learning techniques over classical ones". This can not be stated based on the results of only one, or two, ML methods.

The main limitation of the study should be resolved before publication.

We agree with the fact that the number of machine learning algorithms should be broader. In the previous version, we published in our open-source code (https://github.com/IA-Cardiologia-husa/LAAC_Thrombus_detection_MLvsClassical) another set of algorithms, which have been added to the manuscript (Table 2 - results).

We considered it appropriate to include a bagging algorithm (random forest) and a boosting algorithm (gradient boosting) to cover two of the most widely used nonlinear methods currently in use [Grinsztajn et al., 2022 - https://arxiv.org/abs/2207.08815]. We have also incorporated one more resampling technique (shuffle split).

Besides, if possible, authors could add some other input variables, such as: clinical data.

Clinical data of these patients included in our study have been already published before in other articles. Therefore, we rather not included them in this one. However, mention about this statement has been included now in the manuscript[1–4].

References:

  1. Faroux, L.; Cruz-González, I.; Arzamendi, D.; Freixa, X.; Nombela-Franco, L.; Peral, V.; Caneiro-Queija, B.; Mangieri, A.; Trejo-Velasco, B.; Asmarats, L.; et al. Effect of Glomerular Filtration Rates on Outcomes Following Percutaneous Left Atrial Appendage Closure. The American Journal of Cardiology 2021, 145, 77–84, doi:10.1016/j.amjcard.2020.12.081.
  2. Faroux, L.; Cruz-González, I.; Arzamendi, D.; Freixa, X.; Nombela-Franco, L.; Peral, V.; Caneiro-Queija, B.; Mangieri, A.; Trejo-Velasco, B.; Asmarats, L.; et al. Short-Term Direct Oral Anticoagulation or Dual Antiplatelet Therapy Following Left Atrial Appendage Closure in Patients with Relative Contraindications to Chronic Anticoagulation Therapy. International Journal of Cardiology 2021, 333, 77–82, doi:10.1016/j.ijcard.2021.02.054.
  3. Mesnier, J.; Cruz-González, I.; Arzamendi, D.; Freixa, X.; Nombela-Franco, L.; Peral, V.; Caneiro-Queija, B.; Mangieri, A.; Trejo-Velasco, B.; Asmarats, L.; et al. Early Discontinuation of Antithrombotic Treatment Following Left Atrial Appendage Closure. The American Journal of Cardiology 2022, 171, 91–98, doi:10.1016/j.amjcard.2022.01.055.
  4. Regueiro, A.; Cruz-Gonzalez, I.; Bethencourt, A.; Nombela-Franco, L.; Champagne, J.; Asmarats, L.; Jiménez-Quevedo, P.; Rodriguez-Gabella, T.; Rama-Merchan, J.C.; Puri, R.; et al. Long-Term Outcomes Following Percutaneous Left Atrial Appendage Closure in Patients with Atrial Fibrillation and Contraindications to Anticoagulation. J Interv Card Electrophysiol 2018, 52, 53–59, doi:10.1007/s10840-018-0356-9.

Reviewer 2 Report

DRT after LAA closure is rare but correlates with an increased risk of thromboembolism. In this paper, the authors compare classical multivariable and machine learning techniques for detecting thrombus after LAA closure. For this analysis, the authors use a multicenter registry including 1,150 consecutive patients who underwent left atrial appendage closure from 5 different international centers.

The authors included a relatively large number of patients in the analysis, but the disadvantage of the analysis is the retrospective nature of the study.

The main limitation of the study was not incorporating other algorithms or resampling systems to compare results. Other limitation was the lack of a structured protocol including postprocedural antithrombotic treatment and scheduled follow up.

The authors found that there is no significant difference in the predictive power of thrombus after atrial appendage closure between the classical multivariable and machine learning methodology. In both cases, the predictive power was low.

This study does not significantly broaden the knowledge of DRT after LAA.

Author Response

REVIEWER #2

  • Does the introduction provide sufficient background and include all relevant references? Can be improved.
  • Is the research design appropriate? Can be improvedN
  • Are the methods adequately described? Can be improved

The introduction has been completed and more references have been added. Further explanations and changes about the research design and methods have been included in the new version

Comments and Suggestions for Authors

DRT after LAA closure is rare but correlates with an increased risk of thromboembolism. In this paper, the authors compare classical multivariable and machine learning techniques for detecting thrombus after LAA closure. For this analysis, the authors use a multicenter registry including 1,150 consecutive patients who underwent left atrial appendage closure from 5 different international centers.

The authors included a relatively large number of patients in the analysis, but the disadvantage of the analysis is the retrospective nature of the study.

As you interestingly remark, the retrospective character of the study confers a limitation. This has now been included in the article in the discussion section as one of its limitations.

The main limitation of the study was not incorporating other algorithms or resampling systems to compare results.

We agree with the fact that the number of machine learning algorithms should be broader. In the previous version, we published in our open source code (https://github.com/IA-Cardiologia-husa/LAAC_Thrombus_detection_MLvsClassical) another set of algorithms, which have been added to the manuscript (Table 2).

We considered it appropriate to include a bagging algorithm (random forest) and a boosting algorithm (gradient boosting) to cover two of the most widely used nonlinear methods currently in use [Grinsztajn et al., 2022 - https://arxiv.org/abs/2207.08815].

We have also incorporated another resampling technique in the methods and results section (Shuffle Split). Although we considered adding the jack-knife resampling method, we believe that this technique may incur overfitting [Faber NK, 2007 – DOI 10.1016/j.aca.2007.10.002].

Other limitation was the lack of a structured protocol including postprocedural antithrombotic treatment and scheduled follow up.

A structured protocol for the postprocedural antithrombotic treatment would be definitively interesting. However, as you probably know, antithrombotic treatment after LAAC is nowadays not yet well defined, and the follow-up took part in five different centers, so it was difficult to agree on all the same antithrombotic treatment. Different real-world data analyses do not agree with the optimal postprocedural treatment[5,6]. Furthermore, different ongoing studies are trying to elucidate which treatment is better to reduce both bleeding and thrombotic events (DAPT vs DOACs for example). Moreover, the indication of LAAC varies depending on the patient, from those with new thrombotic events despite oral anticoagulation to those with high-risk bleeding events where anticoagulation is completely forbidden. Therefore, these are the main reasons why there is no structured protocol. By the way, discharge treatment was used in our models to try to find out if any of these possible medications could be determinants for DRT prediction. All these considerations have been included in the new version of the manuscript.

The authors found that there is no significant difference in the predictive power of thrombus after atrial appendage closure between the classical multivariable and machine learning methodology. In both cases, the predictive power was low.

This study does not significantly broaden the knowledge of DRT after LAA.

Despite finding no significant differences in the predictive power of DRT between the classical and the machine learning methodology, we think that these results should be published because of their interest. DRT after LAAC is an important problem due to its clinical impact. Moreover, even though the machine learning methodology used a broader set of variables, there were no differences. This cloud be interesting because not always machine learning can solve the problems we face in medicine. By the way, if machine learning is not capable of predicting DRT there is something we still do not know about DRT. That is why several studies are trying to predict them with heterogeneous results. However, using higher databases or different variables could improve their prediction power of them, leading to selecting those patients who may benefit from an intensive follow-up. These are the main reasons we think this article contributes to increasing the knowledge in DRT after LAAC with the results found.

References:

  1. Freeman, J.V.; Higgins, A.Y.; Wang, Y.; Du, C.; Friedman, D.J.; Daimee, U.A.; Minges, K.E.; Pereira, L.; Goldsweig, A.M.; Price, M.J.; et al. Antithrombotic Therapy After Left Atrial Appendage Occlusion in Patients With Atrial Fibrillation. Journal of the American College of Cardiology 2022, 79, 1785–1798, doi:10.1016/j.jacc.2022.02.047.
  2. Patti, G.; Sticchi, A.; Verolino, G.; Pasceri, V.; Vizzi, V.; Brscic, E.; Casu, G.; Golino, P.; Russo, V.; Rapacciuolo, A.; et al. Safety and Efficacy of Single Versus Dual Antiplatelet Therapy After Left Atrial Appendage Occlusion. The American Journal of Cardiology 2020, 134, 83–90, doi:10.1016/j.amjcard.2020.08.013.

Reviewer 3 Report

This is a multicenter registry including 1,150 consecutive patients who underwent percutaneous left atrial appendage (LAA) closure.  There are two aims of this study.  One is to determine whether any differences exist with respect to the predictive power between the classical and the machine learning methodology to predict device-related thrombus (DRT) after LAA closure.  The other one is to extract the predictor variables of both methodologies and analyze their differences.  DRT was found in 4.3% of patients in this study.  There was no significant difference in the predictive power of DRT between the classical multivariable and machine learning methodology.  The authors could not identify significant predictive parameter relating to DRT, because DRT might be a multifactorial phenomenon.  There were differences between the predictor variable extracted by the two methodologies.  The machine learning methodology utilized several variables to fit its models.

This article was written well and easy to read.  There are couple of points to be noted.

The authors declared no significant differences between the two methodologies were found in the line 278-279.  I would like the authors to explain how to translate this result.  Is there a cutoff point to judge significant differences or not?

Author Response

REVIEWER #3

  • Are the methods adequately described? Can be improved
  • Are the results clearly presented? Can be improved

Further explanation and changes about the methods and the results have been included in the new version.

Comments and Suggestions for Authors

This is a multicenter registry including 1,150 consecutive patients who underwent percutaneous left atrial appendage (LAA) closure.  There are two aims of this study.  One is to determine whether any differences exist with respect to the predictive power between the classical and the machine learning methodology to predict device-related thrombus (DRT) after LAA closure.  The other one is to extract the predictor variables of both methodologies and analyze their differences.  DRT was found in 4.3% of patients in this study.  There was no significant difference in the predictive power of DRT between the classical multivariable and machine learning methodology.  The authors could not identify significant predictive parameter relating to DRT, because DRT might be a multifactorial phenomenon.  There were differences between the predictor variable extracted by the two methodologies.  The machine learning methodology utilized several variables to fit its models.

This article was written well and easy to read.  There are couple of points to be noted.

The authors declared no significant differences between the two methodologies were found in the line 278-279.  I would like the authors to explain how to translate this result.  Is there a cutoff point to judge significant differences or not?

We appreciate the question because we had not in fact reported an objective way of assessing this difference beyond the purely descriptive.

In order to evaluate their significance, we first performed a Levene test between the ratios of the multivariate methodology and the machine learning methodology to evaluate homoscedasticity. The result of the test concluded that the variances of both distributions were different and therefore we performed a Mann-Whitney U test whose final result was not significant.

The results obtained in the Levene and Mann-Whitney U test have been incorporated into the article (lines 323-327).

Round 2

Reviewer 1 Report

The authors improved the paper.